# Effect of Silica Fume on the Rheological Properties of Cement Paste with Ultra-Low Water Binder Ratio

**DOI:** 10.3390/ma15020554

**Published:** 2022-01-12

**Authors:** Juan He, Congmi Cheng, Xiaofen Zhu, Xiaosen Li

**Affiliations:** School of Civil Engineering, Guangzhou University, Guangzhou 510006, China; cehejuan@gzhu.edu.cn (J.H.); zhuxf@pearl-river.com (X.Z.); 2111913138@e.gzhu.edu.cn (X.L.)

**Keywords:** silica fume, rheology, plastic viscosity, yield stress, water binder ratio, hysteresis loop area

## Abstract

The effect of silica fume on the rheological properties of a cement–silica fume–high range water reducer–water mixture with ultra-low water binder ratio (CSHWM) was studied. The results indicate that the W/B ratio and silica fume content have different effects on the rheological parameters, including the yield stress, plastic viscosity, and hysteresis loop area. The shear-thickening influence of CSHWM decreased with the increased silica fume content. When the silica fume content increased from 0% to 35%, the mixture with W/B ratio of 0.19 and 0.23 changed from a dilatant fluid to a Newtonian fluid, and then to a pseudoplastic fluid. When the silica fume content was less than 15%, the yield stress was close to 0. With the increase of silica fume content, the yield stress increased rapidly. The plastic viscosity and hysteresis loop area decreased slightly with the addition of a small amount of silica fume, but increased significantly with the continuous increase of silica fume. Compared with the Bingham and modified Bingham models, the Herschel–Buckley model is more applicable for this CSHWM.

## 1. Introduction

With the use of silica fume and high-range water reducer (HRWR), fresh cementitious composites still have good fluidity at an ultra-low water/binder (W/B) ratio. Therefore, the ultra-high performance concrete (UHPC) has specific properties. In addition to structural applications, UHPC can also be applied to furniture, washbasins, lamps, and art decorations (Figure 1). Cement–silica fume–high range water reducer–water mixture with ultra-low water binder ratio (CSHWM) is the key cementitious component of UHPC.

The silica fume and HRWR strongly affect the rheology of fresh cementitious composites and are vital for the preparation of UHPC [1]. Compared with slump and slump fluidity tests, rheological tests can be used to more effectively analyze the workability of high-performance concrete [2].

Silica fume is a type of pozzolanic material with very small particle size that is widely used in China. Researchers have investigated the effect of silica fume on the rheology of cementitious mixtures [3,4,5,6]. There are several parameters of rheology, and the introduction of silica fume does not affect them all in the same way [7]. Yun et al. [8] found that the addition of silica fume led to a remarkable increase in flow resistance while slightly reducing the torque viscosity. The introduction of silica fume increased the yield stress of fresh cemented tailings backfill but decreased the viscosity [9]. For economic and technical reasons, the silica fume content in most concrete is less than 10%; however, that used in UHPC is much higher. The rheological properties of cement composites with ultra-high silica fume content have not been fully studied.

A rheological curve of cement-based composites is not linear. Therefore, various nonlinear rheological models were used to analyze the rheological properties of cement-based composites [10]. Although the modified Bingham and Herschel–Buckley models have been widely accepted, a negative yield stress may be obtained using these nonlinear rheological models, and the yield stress is also lower. Therefore, Koutný et al. [11] suggested that only the Bingham model be used when analyzing the yield stress of cementitious composites. In addition to the commonly used rheological models, researchers have also proposed new rheological models to describe the rheological behavior of mixtures containing nanoparticles [12]. It should be noted that the models are not completely conflicting. Faraj et al. [13] reported that both the Herschel–Bulkley model and modified Bingham model can be used to analyze the rheological properties of self-compacting high-strength concrete.

Most of the published literature on the rheological properties of concrete has used concrete containing stones and sand. However, the effect of each component on the rheological properties of concrete is complicated. To grasp the rheological properties of fresh UHPC, CSHWM was used in the current study. After eliminating the unstable aggregates, the effect of the W/B ratio and silica fume on the rheology of the stable cement paste was investigated. It provides theoretical guidance for the selection of silica fume content in UHPC mix design.

## 2. Materials and Experiment

### 2.1. Raw Materials

Portland cement of P·I 42.5 produced by Fushun Cement Co., Ltd., Fushun, China. was used. The ingredients of the cement were obtained in accordance with Appendix A of GB (2008) [14]. The chemical contents of Portland cement and silica fume are shown in Table 1.

The content of silicon dioxide in the silica fume was 95.7%. Figure 2 shows the morphology of the Portland cement and silica fume. The ultra-fine silica fume particles were perfectly spherical, whereas the cement particles were irregular with sharp edges and corners. The silica fume was much finer than the cement. The particle sizes of the silica fume and cement were approximately 0.2 and 10 μm, respectively; the densities were 2.21 and 3.11 g/cm^3^, respectively; and the specific surface areas were 2.0 × 10^4^ and 382 m^2^/kg, respectively.

The HRWR used in the study was polycarboxylate with a water reducing rate of 35% and a solid content of 40%.

### 2.2. Mixture Proportion

To grasp the effect of silica fume on the rheology of CSHWMs, CSHWMs with W/B ratios of 0.19 and 0.23 and silica fume contents of 0–35% were employed in this study. The silica fume content was calculated from the quality of the cementitious components. The CSHWMs had a fixed HRWR content of 1.0%. Eight silica fume contents were employed, varying from 0 to 35%. The eight CSHWMs containing 0, 5%, 10%, 15%, 20%, 25%, 30%, and 35% silica fume with a W/B ratio of 0.19 were designated as SF0.19-0, SF0.19-5, SF0.19-10, SF0.19-15, SF0.19-20, SF0.19-25, SF0.19-30, and SF0.19-35, respectively. Those with a W/B ratio of 0.23 were designated as SF0.23-0, SF0.23-5, SF0.23-10, SF0.23-15, SF0.23-20, SF0.23-25, SF0.23-30, and SF0.23-35, respectively. Table 2 shows the typical mixture proportions of two different W/B ratios.

Reducing the W/B ratio is an important way to improve the strength of cement-based materials. However, too low of a W/B ratio reduces the fluidity of mixture, which affects its workability and product performance. To further grasp the effect of the W/B ratio on the rheology of CSHWMs, different W/B ratio mixtures were employed for the rheological tests. The silica fume and HRWR contents were 25% and 1.0%, respectively. Eight W/B ratios of 0.16–0.23 were used. The eight CSHWMs incorporating W/B ratios of 0.16, 0.17, 0.18, 0.19, 0.20, 0.21, 0.22, and 0.23 were designated as SF0.16, SF0.17, SF0.18, SF0.19, SF0.20, SF0.21, SF0.22, and SF0.23, respectively.

### 2.3. Rheological Test

A Brookfield RST-CC rheometer was used for the rheological tests. The outer cylinder radius of the rheometer is 3.0 cm, the inner rotor radius is 1.5 cm, the rotor height is 6.0 cm, the viscosity value range is 5.00 × 10^−5^ to 5.41 × 10^6^ Pa/s, and the rotating speed is 0.01 to 1300 RPM. After mixing, the CSHWM was added to the rheometer until the paddle was completely submerged.

To obtain the stable shear state, the test consisted of two stages: the pre-shear and data-acquisition stages. Before the pre-shear process, the CSHWM was placed in the container for 30 s. During the pre-shear process, the shear speed increased from 0 to 25 r/s within 30 s, remained constant at 25 r/s for 30 s, and then decreased to 0 r/s within 30 s. After resting in the cup for another 30 s, the data-acquisition stage began. The shear speed increased from 0 to 25 r/s within 30 s and then decreased to 0 r/s within 30 s immediately. Each test lasted no more than 10 min. In addition, 250 data points were measured, and the loading and unloading curves were recorded automatically. The mechanism of the rheological test is shown in Figure 3.

## 3. Results and Discussion

### 3.1. Effect of Silica Fume Content on Rheological Properties and Rheological Model of CSHWMs

#### 3.1.1. Rheological Curves of CSHWMs with Different Silica Fume Contents

The effect of the silica fume content on the rheological properties of CSHWMs is well reflected by the rheological curves. The loading and unloading curves of the rheological tests with W/B ratios of 0.19 and 0.23 and silica fume contents ranging from 0 to 35% are presented in Figure 4 and Figure 5. The unloading curve of each CSHWM is below the loading curve, forming a hysteresis loop. This is due to the destruction of the flocculation structure and homogenization of solid particles in the CSHWM under the action of stirring during the rheological test, which resulted in a lower shear stress in the unloading process than in the loading process at the same shear rate [15].

As observed in Figure 4 and Figure 5, the CSHWM exhibited significantly different rheological properties for different silica fume contents. However, the effect of the W/B ratio on the rheological curve is not obvious. When the silica fume content was below 10%, the apparent viscosity of the CSHWM increased with an increasing shear rate, with characteristics of a dilatant-flow fluid with shear thickening (Figure 4a and Figure 5a). As the silica fume content increased, the shear-thickening effect decreased [16]. When the silica fume content was 10–20%, the shear stress increased uniformly with an increasing shear rate, and the apparent viscosity changed slightly, with characteristics of a typical Newtonian fluid (Figure 4b and Figure 5b). When the silica fume content was greater than 20%, the apparent viscosity decreased with increasing shear, with characteristics of a pseudoplastic fluid with shear thinning (Figure 4c,d and Figure 5c,d).

#### 3.1.2. Rheological Model of CSHWMs with Different Silica Fume Contents

The rheological models used in cement-based mixtures include the linear Bingham, modified Bingham, and Herschel–Bulkley models. Some researchers consider fresh cement-based composites to be a type of plastic fluid and adopt the Bingham model for analysis [17,18]. However, other researchers do not find the Bingham model to be feasible [12,19]. To understand the effect of these models on the rheological properties of CSHWMs, the rheological curves of the CSHWMs were analyzed using the above three models. Because the unloading curve is more reliable and repeatable, it was selected for the rheological analysis [20,21]. The fitting results of the rheological parameters are presented in Table 3.

As shown in Table 3, the test data were fitted to the three models. The results indicate that the correlation coefficients (*R*^2^) of the different models were greater than 0.95, which indicates that the three models are appropriate for CSHWMs. Among them, the performance of the Bingham model was slightly worse; the minimum *R*^2^ was 0.9510, and the average *R*^2^ was 0.9870. The Herschel–Bulkley model had the best fitting effect; the minimum *R*^2^ was 0.99957, and the average *R*^2^ was 0.9999. The regression analysis of the rheological data indicates that the *R*^2^ values of CSHWMs with a W/B ratio of 0.19 and 0.23 and silica fume content of 0–35% are close to 1.00, suggesting that the Herschel–Bulkley model can well simulate the rheological properties of CSHWMs.

#### 3.1.3. Yield Stress of CSHWMs with Different Silica Fume Contents

The rheological parameters, including the plastic viscosity, yield stress, and hysteretic loop, were analyzed in this study. For the Herschel–Bulkley model, there is a rheological rate index that can reflect the change of apparent viscosity. The friction between particles, rigid connections between solid particles, and colloidal interaction are factors contributing to the yield stress [22]. The yield stress is the minimum stress at which the paste begins to flow and can be used to characterize the stability of cement paste. Its value depends on the roughness and size of solid particles, particle spacing, filler volume fraction, and interactions between particles, among other properties. The effect of HRWR is also critical.

After obtaining the rheological curve of the CSHWM in steady flow, the yield stress was analyzed using the Bingham, modified Bingham, and Herschel–Bulkley models. Figure 6 shows the effect of the silica fume content on the CSHWM yield stress fitted using the above three models for W/B ratios of 0.19 and 0.23, respectively. For W/B ratios of 0.19 and 0.23 and silica fume contents of 15 wt% or below, the fitting results of the three models all indicated that the yield stress was close to 0.

When the silica fume increased to above 15%, the yield stress rapidly increased. The addition of ultra-fine silica fume has two opposite effects on the CSHWM. On the one hand, the water between the cement particles is extruded by the ultra-fine silica fume, which increases the amount of water wrapping the solid particles, thereby decreasing the yield stress of the paste [23]. On the other hand, a higher content of ultra-fine silica fume results in a larger total specific surface area of the particles, greater formation of the flocculation structure, and a thinner water film on the particle surface. At this time, the cohesion force is very large, and the relative slip resistance between particles is large under the shear force, thereby increasing the yield stress [24]. When silica fume content does not exceed 15%, the two effects counteract each other, and the change of the silica fume content does not cause a significant change in the yield stress. When the silica fume content exceeds 15%, however, increasing the silica fume content is useless to fill the voids of cement particles. Accordingly, the yield stress of the CSHWM greatly increases with increasing silica fume content.

The yield stress values analyzed using the various rheological models differed. The rheological curve is nonlinear in the low-shear-rate region. For a selected model, the accuracy of the shear stress depends on its fitting ability in this region. When the silica fume content is lower than 10% or higher than 15%, CSHWMs exhibit dilatancy and pseudoplasticity, respectively. In this case, the linear Bingham model cannot accurately fit the low-shear-rate region of the nonlinear rheological curve. As a result, the Herschel–Bulkley model and modified Bingham model have larger error in predicting the yield stress than the Bingham model [25]. When the silica fume content is no more than 15%, similar to findings in the literature [19], a negative yield stress is obtained using the Bingham model, which is inconsistent with its physical significance. Although the yield stress obtained from the Herschel–Bulkley model also has a negative value, the deviation from the axis is small. However, the Herschel–Bulkley model provides some advantages in describing the flow behavior of CSHWMs.

When the silica fume content was higher than 20%, the yield stress values predicted from the above three models differed. Accordingly, the estimates of the yield stresses are roughly as follows: Bingham model > modified Bingham model > Herschel–Bulkley model, which is similar to previous findings in the literature [25]. For higher silica fume contents, the deviation of the fitting value is greater.

#### 3.1.4. Plastic Viscosity of CSHWMs with Different Silica Fume Contents

When the paste is in a stable shear state, the ratio of the shear stress to shear rate is the plastic viscosity. Colloidal particle interaction forces, Brownian forces, hydrodynamic forces, and viscous forces between particles all affect the plastic viscosity [22,26]. As an index of flow resistance in cement paste, the plastic viscosity is mainly related to the particle morphology, particle size, and W/B ratio. It can be used to evaluate the compactibility, processability, and segregation resistance.

The Herschel–Bulkley model cannot give the plastic viscosity directly. Ferraris and de Larrard [27] derived an empirical formula (1) for plastic viscosity (μ) from a large number of experiments:(1)$$\mu =\frac{3m}{n+2}\gamma_{max}{}^{n-1}$$
where $\gamma_{max}$ is the maximum shear rate, s^−1^; *m* is the consistency index, Pa·s^−n^; and *n* is the rheological rate index, unitless.

Figure 7 shows the effect of the silica fume content on the plastic viscosity of CSHWMs with W/B ratios of 0.19 and 0.23 according to the Bingham, modified Bingham, and Herschel–Bulkley models. Similar to the results in the literature [28], with increasing silica fume content, the plastic viscosity first decreased slightly and then increased significantly.

The ultra-fine silica fume has high chemical activity, increasing the water demand of the paste [29,30]. These rheological values were obtained with CSHWMs in which the cement and water were combined for less than 10 min; thus, the hydration of cementitious materials is not the main reason for the change of rheological properties. Instead, the ultra-fine particle size and spherical morphology of silica fume may be the best explanation for this behavior [7]. The increase of packing density also increases the total surface area of the CSHWM, where the effects on fluidity are opposite. Therefore, increasing the packing density may increase or decrease the fluidity, depending on the relative magnitudes of such opposite effects [31]. At a relatively low silica fume content, the direct friction between cement particles determines the rheological behavior of the CSHWM. As expected, a small amount of silica fume can significantly increase the filling density of cement. Spherical silica fume can play a good role in the lubrication between cement particles, and the plastic viscosity of the CSHWM decreased slightly. However, when the silica fume content was too high, the total specific surface area of solids increased and more flocculation structures were formed. In addition, the water film on the surface of solid particles was thinner, the flow resistance was larger, and the CSHWM became more viscous [32].

The value of plastic viscosity fitted by the Bingham, modified Bingham, and Herschel–Bulkley models reasonably reflects the effect of the silica fume content on the plastic viscosity of the CSHWM. Similar to the yield stress, the values of plastic viscosity from different models differed. For W/B ratios of both 0.19 and 0.23, the plastic viscosity values fitted by the Bingham model were higher than those fitted by the other two models when the silica fume content was low but lower when the silica fume content was high. When the silica fume content was higher than 10%, the plastic viscosity values fitted by the modified Bingham model were very close to those fitted by the Herschel–Bulkley model. However, when the silica fume content was less than 10%, the plastic viscosity of the modified Bingham model was lower. Considering the variation of plastic viscosity and yield stress, the Herschel–Bulkley model is suitable for the rheological analysis of CSHWMs. It should be noted that, although the rheological curves were in good agreement with the Herschel–Bulkley model, there are still shortcomings in using this model [21]. First, although it is superior to the other two models, a yield stress of slightly less than 0 might still be obtained during the fitting process. There is no reasonable physical explanation for this result. Therefore, the yield stress must be limited to 0 Pa. Second, the plastic viscosity of the Herschel–Bulkley model is derived from an empirical formula, and its physical significance is not clear.

#### 3.1.5. Rate Index of CSHWMs with Different Silica Fume Contents

The Herschel–Bulkley model has a special advantage: there is a rate index (*n*) that reflects the variation of the apparent viscosity with the shear rate. A rate index of less than 1.0 indicates that the apparent viscosity decreases with increasing shear rate and the CSHWM has shear-thinning characteristics. A rate index greater than 1.0 indicates that the apparent viscosity decreases with decreasing shear rate, and the CSHWM has shear-thickening characteristics. When *n* = 1, the apparent viscosity remains constant as the shear rate increases, and the model is simplified to a linear Bingham model.

Figure 8 shows the relationship between the rate index (*n*) and silica fume content for W/B ratios of 0.23 and 0.19.

As observed in Figure 8, *n* decreases with increasing silica fume content. When the silica fume content is less than 10%, *n* > 1, the CSHWM is a dilatant fluid with shear-thickening characteristics. When the silica fume content is greater than 15%, *n* < 1, the CSHWM is a pseudoplastic fluid with shear-thinning characteristics. When the silica fume content is 10–15%, n≈1, the yield stress is close to 0, and the CSHWM behaves as a linear Newtonian fluid. The rheological properties reflected by the rate index of the CSHWM are consistent with the rheological curves presented in Figure 4 and Figure 5. When the silica fume content is low, it is mainly used for filling rather than dispersing cement particles. At higher shear rates, the contact friction between irregular cement particles is the main reason for the increase of apparent viscosity, and the rate index (*n*) is greater than 1.0. At lower silica fume content, the contact friction resistance between cement particles is higher and *n* is higher. With increasing silica fume content, the large number of ultra-fine spherical silica fume particles increases the dispersion of cement particles, thus decreasing the shear-thickening behavior. When the silica fume content is greater than 15%, the cement particles are fully dispersed, and the friction resistance between irregular cement particles decreases in the shearing process. The large amount of ultra-fine silica fume causes the formation of more flocculating structures in the CSHWM. With increasing shear rate, the flocculation structure is continuously destroyed. The primary reason for shear thinning is that the friction bond between particles in the undisturbed structure gradually breaks [11].

#### 3.1.6. Hysteresis Loop Area of CSHWMs with Different Silica Fume Contents

When the shear rate of a CSHWM increases continuously from 0 to a constant value and then decreases gradually from this constant value to 0, the closed shear stress–shear rate curve is a hysteresis loop. The rheological curve forms a hysteresis loop, which indicates that the flocculation structure of the CSHWM is broken during the test [33]. The area of the hysteresis loop represents the amount of the flocculation structure that hinders the flow of CSHWM and the energy required to break the flocculation structure in the shearing [34]. A larger hysteresis loop area indicates more flocculating structures and a greater required energy to break the structure.

Figure 9 shows the relationship between the hysteresis loop area and the silica fume content of the CSHWM. As the silica fume content increased, the area of the hysteresis loop first decreased and then increased. The decrease of the hysteresis loop area is due to a small amount of silica fume filling the gaps between cement particles. The released water destroyed the flocculation structure of the CSHWM and caused more water release, leading to the decrease of the hysteresis loop area. When the silica fume content continued to increase, a large amount of silica fume itself formed more flocculation structures, hindering the flow of the CSHWM and resulting in the increase of the hysteresis loop area.

### 3.2. Effect of W/B Ratio on Rheological Properties and Rheological Model of CSHWMs

#### 3.2.1. Rheological Curves and Rheological Models of CSHWMs with Different W/B Ratios

With increasing W/B ratio, the plastic viscosity of cement paste can be reduced; however, it also causes other problems, such as bleeding or strength decrease. Because of workability and mechanical properties, the lower W/B ratio of UHPC fluctuates in a certain range. The W/B ratio affects the rheology of cement paste by affecting the thickness of the water film [32]. The rheological curves can reflect this effect intuitively. Figure 10 presents the rheological curves of CSHWMs with W/B ratios of 0.16 to 0.23. The apparent viscosity of each CSHWM decreases as the shear rate increases, characteristic of a pseudoplastic fluid with shear-thinning properties. Figure 10 also shows that a larger W/B ratio results in a smaller shear stress at the same shear rate. This is because a larger W/B ratio results in a thicker water film on the surface of solid particles and less relative movement resistance between particles.

Similar to the experiments with different silica fume contents, the rheological curves of the CSHWM were analyzed using the Bingham, modified Bingham, and Herschel–Bulkley models. Table 4 presents the fitting results of the rheological parameters. The minimum correlation coefficient (*R*^2^) fitted by each model for the rheological curves was 0.98973, further indicating that the models can all be well used for CSHWMs. Among the three models, the Herschel–Bulkley model had the best fitting effect, with a minimum *R*^2^ of 0.99942 and average *R*^2^ of 0.99978. This means that the Herschel–Bulkley model can well simulate the rheological properties of CSHWMs with W/B ratios of 0.19 to 0.23 and a silica fume content of 25%.

#### 3.2.2. Plastic Viscosity and Yield Stress of CSHWMs with Different W/B Ratios

Figure 11 shows the relationship between the plastic viscosity, yield stress, and W/B ratio fitted using the three models.

As the W/B ratio increases, both the plastic viscosity and yield stress decrease. With increasing W/B ratio, the thickness of the water film on the surface of solid particles increased, which led to a change of the flocculation structure and friction between particles. Both of these changes affected the rheological properties of the mixture [35]. When the W/B ratio of the mixture was low, the silica fume and cement particles were more likely to agglomerate and adhere to each other under the action of van der Waals and electrostatic forces, forming flocculation structures [15]. This resulted in an increase in the resistance to the flow of the CSHWM. Therefore, a larger W/B ratio resulted in lower plastic viscosity and yield stress. Increasing the W/B ratio led to an increase in the distance and amount of free water between particles, the lubrication of free water, and a decrease in the friction and adhesion between particles. Macroscopically, the plastic viscosity and yield stress of the CSHWM gradually decreased. The plastic viscosity and yield stress differed for the same CSHWM fitted by the different rheological models. The fitting results of the yield stress were roughly as follows: Bingham model > modified Bingham model > HB model. The plastic viscosities analyzed by the modified Bingham and Herschel–Bulkley models were similar, with both values being significantly higher than that of the Bingham model.

Yammine et al. [36] showed that a strong transition exists in the rheological behavior of fresh concrete between a regime dominated by the friction between aggregate particles and a regime dominated by hydrodynamic interactions. A similar transition also exists in CSHWMs. When the W/B ratio is low, the direct friction between solid particles has a greater effect on the rheological behavior of the CSHWM. When the W/B ratio is high, the solid particles are coated with a thick water film, and the hydrodynamic interactions have a greater effect on the rheological behavior of the CSHWM. The effect of the water film thickness on the yield strength and plastic viscosity is smaller than that of inter-particle friction. As shown in Figure 11, with increasing W/B ratio, the plastic viscosity and yield stress rapidly decreased.

#### 3.2.3. Hysteresis Loop Area of CSHWMs with Different W/B Ratios

As observed in the rheological curves in Figure 12, the hysteresis loop area gradually decreased with increasing W/B ratio. When the W/C ratio was low, the flocculation structure more easily formed in the CSHWM [33]. The yield stress, as a characteristic of the resistance to flow in CSHWM, was affected by the destruction degree of the flocculation structure. For larger W/B ratios, less flocculation structure was present. The continuous shear destroyed the flocculation structure and released the water. Accordingly, the shear stress decreased continuously. At the same shear rate, the yield stress of the unloading curve was lower than that of the loading curve. When more flocculating structures were present, the distance between the unloading and loading curve was greater, and the hysteretic loop area was larger.

## 4. Conclusions

The effect of the silica fume content and W/B ratio on the rheological properties of CSHWMs was investigated, and the following conclusions can be drawn:
When the silica fume content was 15% or below, the change of the silica fume content did not cause a significant change of the yield stress. When the silica fume content was higher than 15%, the yield stress increased sharply as the silica fume content increased. With increasing silica fume content, the plastic viscosity and hysteresis loop area first decreased and then increased.The effect of the water-film thickness on the yield strength and plastic viscosity was smaller than that of friction. As the W/B ratio increased, the plastic viscosity and yield stress rapidly decreased.Among the linear Bingham, modified Bingham, and Herschel–Bulkley models, the Herschel–Bulkley model is the most suitable for CSHWMs. The shear-thickening behavior decreased with increasing silica fume content. When the silica fume content was less than 10%, the CSHWM with a W/B ratio of 0.19 or 0.23 was a dilatant fluid with shear-thickening characteristics. When the silica fume content was greater than 15%, the CSHWM was a pseudoplastic fluid with shear-thinning characteristics. When the silica fume content was 10–15%, the yield stress was close to 0, and the CSHWM was a linear Newtonian fluid.The values of the yield stress and plastic viscosity fitted by different rheological models differed for the same CSHWM.

## Figures and Tables

**Figure 1 materials-15-00554-f001:**
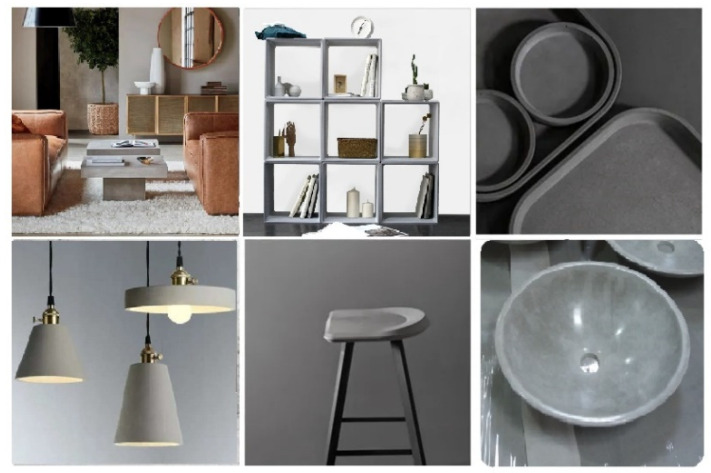
The products of UHPC.

**Figure 2 materials-15-00554-f002:**
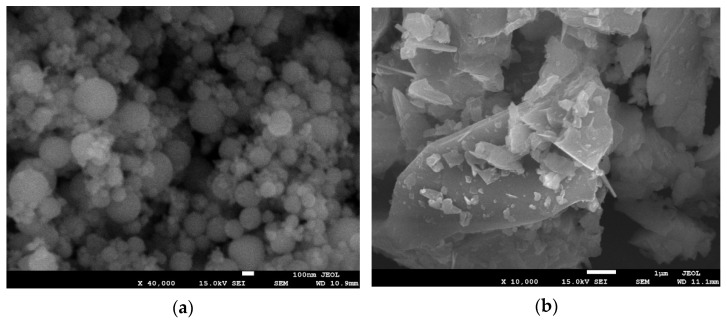
SEM images of silica fume and cement, (**a**) silica fume and (**b**) cement.

**Figure 3 materials-15-00554-f003:**
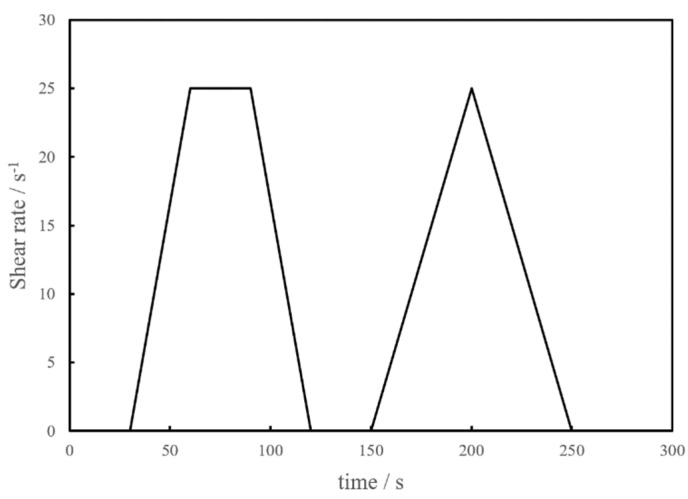
The rheological mechanism.

**Figure 4 materials-15-00554-f004:**
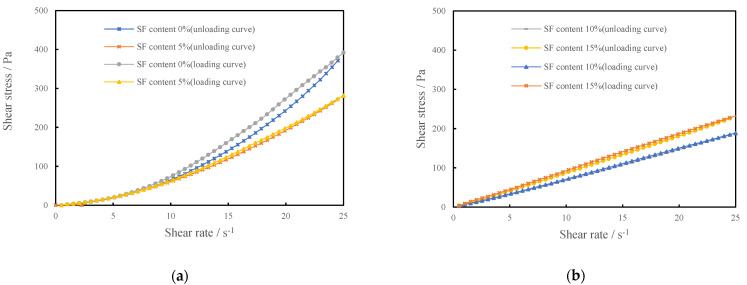
The rheological curve of CSHWMs (W/B = 0.19), (**a**) SF0.19-0% and SF0.19-5, (**b**) SF0.19-10% and SF0.19-15, (**c**) S SF0.19-20 and SF0.19-25, (**d**) SF0.19-30 and SF0.19-35.

**Figure 5 materials-15-00554-f005:**
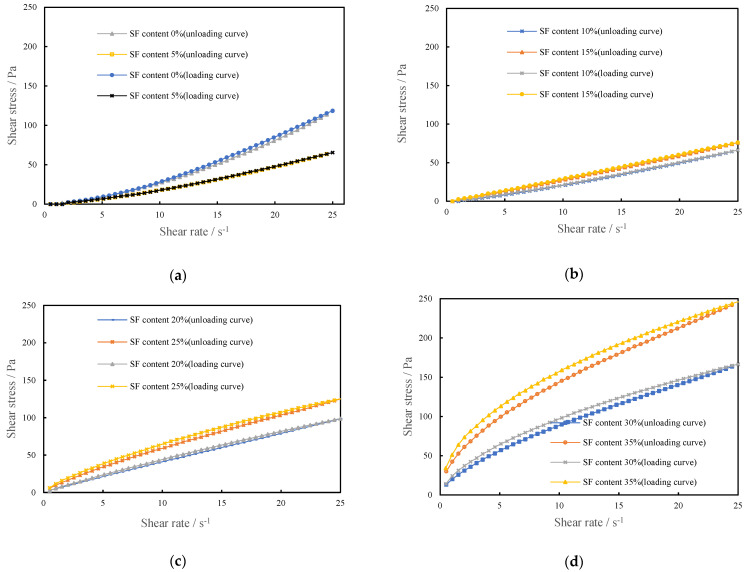
The rheological curve of CSHWMs (W/B = 0.23), (**a**) SF0.23-0% and SF0.23-5, (**b**) SF0.23-10% and SF0.23-15, (**c**) S SF0.23-20 and SF0.23-25, (**d**) SF0.23-30 and SF0.23-35.

**Figure 6 materials-15-00554-f006:**
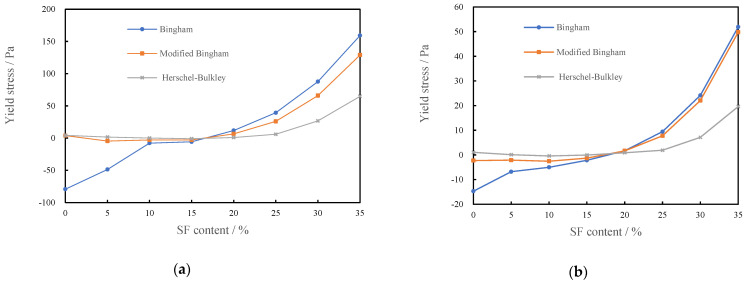
The relationship between yield stress and silica fume content of CSHWM, (**a**) W/B = 0.19, (**b**) W/B = 0.23.

**Figure 7 materials-15-00554-f007:**
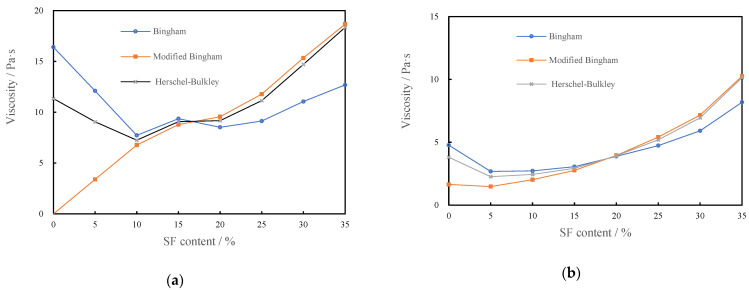
The relationship between the plastic viscosity and silica fume content, (**a**) W/B = 0.19, (**b**) W/B = 0.23.

**Figure 8 materials-15-00554-f008:**
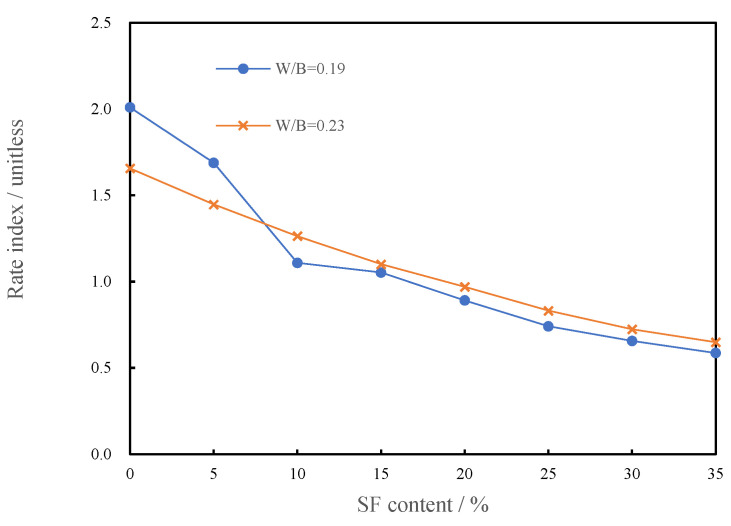
The relationship between the rate index and silica fume content.

**Figure 9 materials-15-00554-f009:**
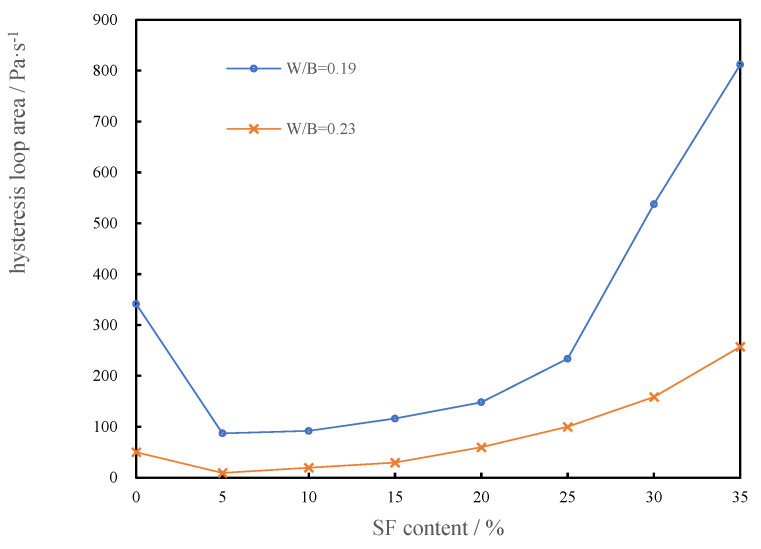
Effect of silica fume content on hysteresis loop area of CSHWM.

**Figure 10 materials-15-00554-f010:**
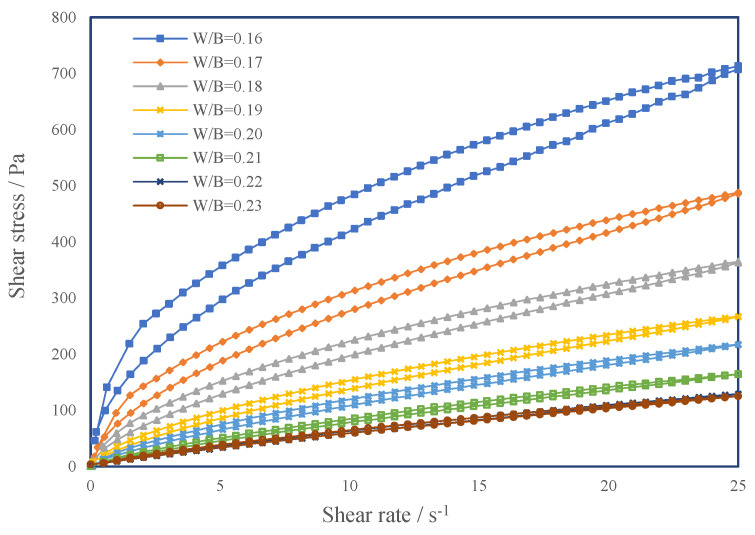
The rheological curves of CSHWMs with different W/B ratios.

**Figure 11 materials-15-00554-f011:**
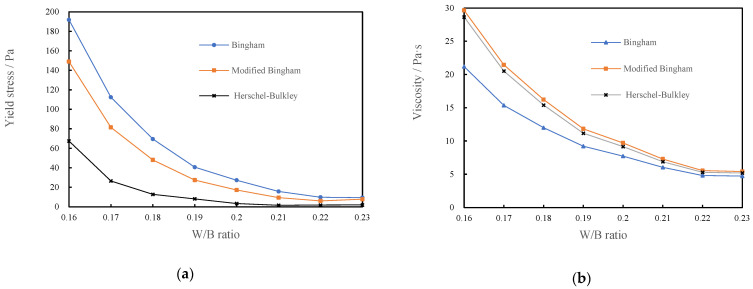
The relationship between the yield stress, plastic viscosity and W/B ratio of CSHWM, (**a**) the yield stress, (**b**) the plastic viscosity.

**Figure 12 materials-15-00554-f012:**
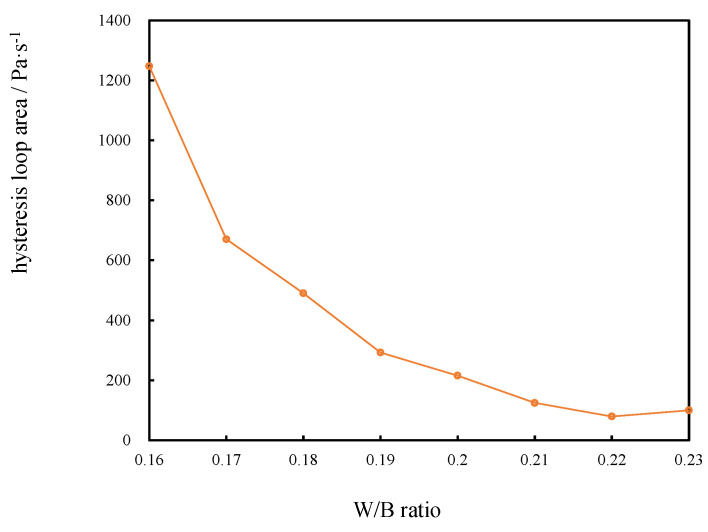
The relationship between the hysteresis loop area and W/B ratio of CSHWM.

**Table 1 materials-15-00554-t001:** CSHWMs of cement and silica fume used.

Composition	SiO_2_	CaO	Al_2_O_3_	Fe_2_O_3_	MgO	SO_3_	NaO_2_/KO_2_	LOI
Cement	20.94	64.02	4.85	3.44	1.70	1.88	0.5	1.88
Silica fume	95.70	0.10	0.06	0.005	0.09	1.03	0.49	2.04

**Table 2 materials-15-00554-t002:** Typical mixture proportions of CSHWM.

Series	SF Content/%	W/B	Mixture Proportion (kg/m^3^)			
			Cement	Silica Fume	Water *	HRMR *
SF0.19-20	20	0.19	800	200	190	10
SF0.23-25	25	0.23	750	250	230	10

Note: Water * includes water in the HRWR; HRWR * denotes the dry extract content of the liquid-based HRWR.

**Table 3 materials-15-00554-t003:** Rheological parameters of CSHWMs with different silica fume contents.

W/B	SF (%)	Bingham Model Y = τ_o_ + μ·X			Modified Bingham Model Y = τ_o_ + μ·X + C·X^2^				Herschel–Bulkley Model Y = τ_o_ + m·X^n^					Hysteresis Loop Area (Pa/s)
		τ_o_ (Pa)	μ (Pa·s)	R^2^	τ_o_ (Pa)	μ (Pa·s)	C (Pa·s^2^)	R^2^	τ_o_ (Pa)	m (Pa·s^n^)	n	μ (Pa·s)	R^2^	
0.19	0	−79.2466	16.4030	0.95097	3.8533	−0.0007	0.61	0.99981	4.2698	0.5884	2.01	11.3399	0.99982	341.2390
	5	−48.6412	12.1000	0.97402	−4.5352	3.3901	0.32	0.99991	1.6574	1.2144	1.69	9.0597	0.99998	87.04490
	10	−7.7685	7.7268	0.99918	−2.9249	6.7703	0.04	0.99997	0.0831	5.2966	1.11	7.2539	0.99998	91.8436
	15	−5.7596	9.3667	0.99978	−2.8114	8.7845	0.02	0.99997	−0.8975	7.7940	1.05	9.0739	0.99996	116.0800
	20	11.7966	8.5223	0.99908	6.5571	9.5571	−0.04	0.99983	0.9649	12.5397	0.89	9.1737	0.99993	148.1030
	25	39.3829	9.1335	0.99474	25.9601	11.7840	−0.10	0.99901	5.9423	23.4052	0.74	11.1434	0.99974	233.7950
	30	87.6637	11.0516	0.99064	65.9564	15.3370	−0.16	0.99823	26.7509	39.3483	0.66	14.7115	0.99964	537.6000
	35	159.2611	12.6876	0.9863	128.9839	18.6650	−0.22	0.99746	64.8383	59.8548	0.59	18.3592	0.99957	811.9190
0.23	0	−14.6917	4.7705	0.96852	−2.2965	1.6473	0.1260	0.99996	1.0606	0.5607	1.66	3.8101	0.99993	50.0400
	5	−6.7971	2.6878	0.98334	−2.1229	1.4772	0.0492	0.99996	0.1087	0.6173	1.45	2.2676	0.99993	9.3500
	10	−5.0183	2.7234	0.99299	−2.5482	2.0233	0.0292	0.99993	−0.3942	1.1371	1.26	2.4423	0.99994	19.4610
	15	−2.1797	3.0577	0.99798	−1.2888	2.7607	0.0129	0.99996	−0.0433	2.1866	1.10	2.9279	0.99993	29.5600
	20	1.6977	3.8796	0.99975	1.6475	3.9553	−0.0038	0.99991	0.8839	4.2850	0.97	3.9294	0.99992	59.6000
	25	9.4499	4.7344	0.99597	7.7533	5.4081	−0.0302	0.99979	1.8923	8.4218	0.83	5.1926	0.99993	99.9600
	30	24.1333	5.9120	0.98598	22.0627	7.1561	−0.0590	0.99922	7.1163	15.3387	0.72	6.9436	0.99978	158.6500
	35	51.9112	8.1884	0.97203	49.7632	10.2565	−0.1020	0.99891	19.6398	27.6977	0.65	10.1438	0.99978	257.3300

**Table 4 materials-15-00554-t004:** Rheological parameters of CSHWMs with different W/B ratios.

W/B	Bingham Model Y = τ_o_+μ·X			Modified Bingham Model Y = τ_o_ + μ·X + C·X^2^				Herschel–Bulkley Model Y = τ_o_ + m·X^n^					Hysteresis Loop Area (Pa/s)
	τ_o_ (Pa)	μ (Pa·s)	R^2^	τ_o_ (Pa)	μ (Pa·s)	C (Pa·s^2^)	R^2^	τ_o_ (Pa)	m (Pa·s^n^)	n	μ (Pa·s)	R^2^	
0.16	191.76344	21.19016	0.98973	148.8529	29.65565	−0.31288	0.99776	67.48353	79.71507	0.64278	28.656856	0.99942	1247.2
0.17	112.33811	15.36048	0.99054	81.4527	21.45612	−0.22535	0.9985	26.47295	55.36596	0.6535	20.519123	0.99977	670.67
0.18	69.46219	11.99865	0.99246	48.03614	16.22866	−0.15639	0.99875	12.75377	37.46153	0.69009	15.406513	0.99976	490.47
0.19	40.62626	9.21337	0.99506	27.38048	11.82912	−0.09672	0.99915	8.07598	23.01547	0.74816	11.169709	0.99981	292.69
0.2	27.23733	7.72087	0.99595	17.18274	9.70635	−0.07342	0.99932	3.34747	17.61791	0.77216	9.1569469	0.99982	215.56
0.21	15.7223	6.03085	0.99741	9.35907	7.28774	−0.04649	0.99963	1.57706	11.64819	0.81692	6.8812974	0.99989	124.91
0.22	9.79596	4.80847	0.9985	6.09555	5.53925	−0.02703	0.99968	1.80666	7.84926	0.8628	5.2888542	0.99987	79.81
0.23	9.4499	4.7344	0.99597	7.7533	5.4081	−0.0302	0.99979	1.8923	8.4218	0.83	5.1926	0.99993	99.96

## Data Availability

Some or all data, models, or code that support the findings of this study are available from the corresponding author upon reasonable request.

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
