# Peer review of "Effect of Silica Fume on the Rheological Properties of Cement Paste with Ultra-Low Water Binder Ratio"

_materials, 2022, doi:10.3390/ma15020554_

Round 1
Reviewer 1 Report
Dear authors,
I would thank you for the nice paper. The paper is well written and well presented. Please just correct :
Line 41 : Rheological curve
Line 276 : As observed in Figure 8
Thank you in advance,
Author Response
Point 1: Line 41: Rheological curve
Response 1: "Theoretical curve" has been revised to "Rheological curve". Please see Page 2 Line 47 of the revised manuscript highlighted in red fonts.
Point 2: Line 276: As observed in Figure 8
Response 1: "As observed in Figure 5" has been revised to " As observed in Figure 8". Please see Page 10 Line 284.
Thank you for your constructive comment.

Reviewer 2 Report
The study investigated the effect of silica fume on the rheological properties of a cement – silica fume. The authors found that W / B ratio and silica fume content had an influence on the plastic viscosity, and hysteresis loop area. Moreover, on the basis of the conducted research, the most adequate model for the tested materials was determined (Herschel-Buckley model). The work is written in understandable language, it is easy to read. Importantly, the authors of the work not only describe the research results but also explain the phenomena that occur. Moreover, the presented results are of an application nature. The work is well thought out, the figures have a uniform structure, which proves that the work has been thoroughly prepared. I believe that the work is suitable for publication in Materials after editing (for example, a mistake in the name of one of the authors).
Author Response
Point: The work is well thought out, the figures have a uniform structure, which proves that the work has been thoroughly prepared. I believe that the work is suitable for publication in Materials after editing (for example, a mistake in the name of one of the authors).
Response: Thank you for your constructive comment.
A mistake of one author’s name has been corrected. “Xiaosen Li” is the correct spelling. Other mistakes were also corrected, such as spelling errors.

Reviewer 3 Report
The following comments and suggestions will improve the material presented.
Review comments
- There are many abbreviations. A nomenclature should be added at the beginning or end of the article.
- The authors should explain what a HRWR is doing in the mixture.
- Lines 120-121 contradict with lines 123-124 because in the first sentence it is written that the loading curve is below the unloading and in the second that the shear stress is lower in the unloading curve. Please explain.
- In Figures 4 and 5 the unloading curve should be presented with different color or dashed line and it should be written in the text which one is the loading and which the unloading curve.
- Can the experimental results be added in Figure 6?
Author Response
Point 1: There are many abbreviations. A nomenclature should be added at the beginning or end of the article.
Response 1: There are four abbreviations of CSHWM, HRWR, W/B and UHPC in the manuscript. According to the reviewer’s suggestion, the abbreviation CSHWM was added in the first paragraph of the manuscript. Please see Page 1 Line 28-29 of the revised manuscript highlighted in red fonts.
Point 2: The authors should explain what a HRWR is doing in the mixture.
Response 2: HRWR is the basic component of UHPC. There have been a lot of studies on the effects of HRWR on UHPC performance. This is not covered in this manuscript, so the role of HRWR in the mixture is not introduced.
Point 3: Lines 120-121 contradict with lines 123-124 because in the first sentence it is written that the loading curve is below the unloading and in the second that the shear stress is lower in the unloading curve. Please explain.
Response 3: It has been revised according to the suggestions of reviewer. Please see Page 4 Line 128-129 of the revised manuscript highlighted in red fonts.
Point 4: In Figures 4 and 5 the unloading curve should be presented with different color or dashed line and it should be written in the text which one is the loading and which the unloading curve.
Response 4: It has been revised according to the suggestions of reviewer. Please see Figures 4 and 5 of the revised manuscript.
Point 5: Can the experimental results be added in Figure 6?
Response 5: Figure 6 is based on the data in Table 3. To understand the effect of different models on the rheological properties of CSHWMs, the rheological curves of the CSHWMs were analyzed using three models. The fitting results of the rheological parameters are presented in Table 3. Please see Table 3 and Figures 6 of the revised manuscript.
Thank you for your constructive comment.
Reviewer 4 Report
The proposed work focuses on the Effect of Silica Fume on the Rheological Properties of Cement 2 Paste with Ultra-Low Water Binder ratio. It is of potential interest to Materials journal readers.
Despite the importance of the subject addressed, this work needs many improvements to be ready for the publication in the Materials journal.
Specific points of improvement :
- Abstract is too short
- Authors affiliations are incomplete and don't meet to the journal requirements.
- Literature review section must be improved by more previous researches.
- The objective of this research must be more developed.
- Quality of figures must be improved.
- Method section must be more developed and the tests description must be more detailed according to the appropriate standard.
- All test standards must be indicated in the manuscript.
- The conclusion section is too long. Only the main results must be indicated.
Author Response
Point 1: Abstract is too short
Response 1: It has been revised according to the suggestions of reviewer. Please see Page 1 Line 8-19 of the revised manuscript highlighted in red fonts.
Point 2: Authors affiliations are incomplete and don't meet to the journal requirements.
Response 2: It has been revised according to the suggestions of reviewer. Please see Page 1 Line 5-7 of the revised manuscript highlighted in red fonts.
Point 3: Literature review section must be improved by more previous researches.
Response 3: It has been revised according to the reviewers' suggestions. Specifically, it includes the definition of abbreviations (Page 1 Line 28-29), misspelling (Page 2 Line 47) and the research purposes (Page 2 Line 64-65).
Point 4: The objective of this research must be more developed.
Response 4: The objective of this research is described in the introduction of the article. And it has been improved according to the reviewer’s suggestion. Please see Page 2 Line 59-65 of the revised manuscript.
Point 5: Quality of figures must be improved.
Response 5: It has been revised according to the suggestions of reviewer. Please see Figures 1 and 2 of the revised manuscript.
Point 6: Method section must be more developed and the tests description must be more detailed according to the appropriate standard. All test standards must be indicated in the manuscript.
Response 6: Common experimental methods were used in this study, so the methods were not described in detail.
Point 7: The conclusion section is too long. Only the main results must be indicated.
Response 7: It has been revised according to the suggestions of reviewer. Please see Page 15 Line 397-406 of the revised manuscript highlighted in red fonts.
Thank you for your constructive comment.

Round 2
Reviewer 4 Report
I think the revised version of the submitted paper is well improved by considering the reviewers comments and recommendations.
Indeed, I think that this paper can be accepted in its present format.